# Exploring Total Immunoglobulin A’s Impact on Non-Biopsy Diagnosis of Celiac Disease: Implications for Diagnostic Accuracy

**DOI:** 10.3390/nu16183195

**Published:** 2024-09-21

**Authors:** Alberto Raiteri, Alessandro Granito, Dante Pio Pallotta, Alice Giamperoli, Agnese Pratelli, Giovanni Monaco, Chiara Faggiano, Francesco Tovoli

**Affiliations:** 1Division of Internal Medicine, Hepatobiliary and Immunoallergic Diseases, IRCCS Azienda Ospedaliero-Universitaria di Bologna, 40138 Bologna, Italy; chiara.faggiano@aosp.bo.it (C.F.); francesco.tovoli@unibo.it (F.T.); 2Department of Medical and Surgical Sciences, University of Bologna, 40138 Bologna, Italy; dantepio.pallotta@studio.unibo.it (D.P.P.); alice.giamperoli@studio.unibo.it (A.G.); agnese.pratelli@studio.unibo.it (A.P.); giovanni.monaco@studio.unibo.it (G.M.)

**Keywords:** celiac disease, intestinal villous atrophy, non-biopsy diagnosis, tissue transglutaminase antibodies, total IgA, diagnostic accuracy

## Abstract

Objective: In the current debate surrounding the biopsy-free diagnosis of CeD, it is crucial to identify factors influencing the accuracy of results. This study investigated the impact of total IgA on the non-invasive diagnosis of celiac disease (CeD). Methods: We retrospectively assessed total IgA titers’ influence on the diagnostic accuracy of different tTG-IgA thresholds compared to the upper reference value (UNL). Results: Of 165 included patients, tTG-IgA values at 10× UNL and 6× UNL showed specificity of 82.6% and 73.9% and sensitivity of 49.3% and 69.0%, respectively, in predicting intestinal villous atrophy (Marsh 3). In 130 patients, total IgA levels were known at baseline. These patients were divided into three tertiles according to total IgA, i.e., patients with lower, intermediate, or higher total IgA within the population. For patients with total IgA ≥ 245 mg/dL, using a tTG-IgA cutoff of 6× UNL instead of 10× UNL resulted in decreased specificity from 71.4% to 42.8% and increased sensitivity from 67.6% to 81.1%. For patients with total IgA < 174 mg/dL and between 174 mg/dL and 245 mg/dL, using a tTG-IgA cutoff of 6× UNL instead of 10× UNL maintained specificity (75.0% and 85.7%, respectively) with increased sensitivity (from 46.2% to 64.1% and from 36.1% to 52.8%, respectively). Conclusions: In conclusion, total IgA influences the diagnostic accuracy of a predetermined tTG-IgA cutoff. Greater consideration should be given to total IgA, beyond its deficiency, in evaluating the applicability and accuracy of non-invasive CeD diagnosis.

## 1. Introduction

Celiac disease (CeD) is a multi-system, immune-mediated disease that primarily affects the small intestine. It is characterized by an abnormal autoimmune response involving T lymphocytes and occurs in genetically predisposed individuals who consume gluten [1]. The clinical presentation of CeD varies greatly and changes over time, so current guidelines recommend testing all patients with classic, non-classic, gastrointestinal, and extra-intestinal symptoms and those at high risk [2,3]. (1) European Society Paediatric Gastroenterology, Hepatology and Nutrition (ESPGHAN) guidelines have allowed a CeD diagnosis in pediatric patients without histological investigation if their immunoglobulin A anti-tissue transglutaminase antibody (tTG-IgA) levels are at least 10 times the upper normal limit, and they have classical symptoms, positive immunoglobulin A anti-endomysial antibodies (EMA-IgA), and a predisposing human leukocyte antigen (HLA) serotype DQ-2 or DQ-8 [4]. The 2020 revision of the ESPGHAN guidelines no longer requires the presence of classical symptoms, EMA-IgA positivity, or predisposing HLA for diagnosis [4]. However, in adults, a diagnosis of CeD without histological confirmation is still not permitted by international guidelines [3]. Nevertheless, since the beginning of the XXI century, the appropriateness of performing a diagnosis of CeD without biopsy has also been widely investigated in adult patients [5]. A tTG-IgA value ≥ 10× UNL in adults, using various diagnostic kits, predicts the presence of intestinal villous atrophy (histological lesion Marsh 3) with sensitivity values between 30.0% and 54% and specificity values between 83% and 100% [6,7]. The values vary mainly according to the population being tested, i.e., whether the test is used in a population with a higher or lower pre-test probability or in the general population.

Over time, increasing attention has been given to the total immunoglobulin A (IgA) levels in patients suspected of having CeD, particularly due to the higher prevalence of CeD in patients with IgA deficiency [8]. IgA deficiency is classified as absolute or selective (IGASD) when total IgA is below 7 mg/dL and as partial (IGAPD) when it is below 2 standard deviations from the normal range but still detectable (7–70 mg/dL) [9]. In IGAPD, a very high percentage of patients produce IgA autoantibodies (such as tTG-IgA), making it possible to detect these antibodies [8,10,11]. Some authors recommend testing for IgA autoantibodies in patients with total IgA levels above 50 mg/dL [12]. However, in patients with IGASD, detecting IgA autoantibodies is not possible, so testing for IgG autoantibodies is necessary [8]. Current guidelines suggest testing serum IgA levels in all patients with suspected CeD [3].

Apart from identifying patients with IgA deficiency, the impact of total IgA levels on the diagnostic accuracy of IgA autoantibodies has been less studied. It has been observed that patients with higher IgA autoantibody levels tend to have higher total serum IgA [13]. Moreover, elevated IgA levels, over 400 mg/dL in adults, have not been extensively studied. However, in pediatric patients, high IgA levels have been associated with conditions such as IgA nephropathy, Henoch–Schönlein purpura, primary immunodeficiency, rheumatological diseases, chronic inflammatory bowel disease, and CeD [14]. In adults, elevated IgA levels may be linked to alcohol consumption, metabolic syndrome, or the presence of a monoclonal component [15].

Starting from the evidence of higher levels of IgA class antibodies in patients with higher titers of total IgA, we investigated the possible influence of total IgA values on the diagnostic accuracy of an established threshold value of tTG-IgA/UNL in predicting the presence of histological abnormalities characteristic of CeD.

## 2. Materials and Methods

### 2.1. Study Design

This was a retrospective observational study. The main aim was to investigate the influence of total IgA values on the diagnostic accuracy of an established threshold value of tTG-IgA/UNL in predicting the presence of intestinal villous atrophy (Marsh 3 histological finding). A secondary objective was to confirm the diagnostic accuracy data of the threshold value of tTG-IgA ≥ 10× UNL in predicting the presence of intestinal villous atrophy (Marsh 3 histological finding).

### 2.2. Study Population

The study included all adult patients, aged over 18, of any gender or ethnicity, who were evaluated for suspected celiac disease at the Centre for the Study and Diagnosis of Coeliac Disease at IRCCS Azienda Ospedaliero-Universitaria di Bologna (Italy) between January 2017 and December 2022. Other inclusion criteria were (1) to be on a gluten-containing diet at the time of the first consultation; (2) availability of a tTG-IgA antibody test at the hospital reference laboratory on a gluten-containing diet; (3) availability of duodenal biopsies performed on a gluten-containing diet; (4) the interval between tTG-IgA titer assessment and duodenal biopsies not exceeding 8 weeks.

### 2.3. Laboratory Investigations

#### 2.3.1. tTG-IgA Antibodies

All patients performed a tTG-IgA antibody determination, analyzed at the reference laboratory of our hospital. tTG-IgA antibodies were measured by an FEIA (Fluoro Enzyme Immuno Assay) ELiA Celikey IgA kit (Phadia AB, Uppsala, Sweden) and expressed as unit per milliliters. The test was defined as positive for a value >7 U/mL and had a maximum value of 128 U/mL.

#### 2.3.2. Total IgA

The total IgA titer was determined at the same time as the tTG-IgA titer as part of the normal clinical practice [3]. Total serum IgA levels were measured by nephelometry and were expressed in mg/dL, with values defined as normal between 70 and 400 mg/dL.

### 2.4. Esophagogastroduodenoscopy (EGDS) and Histological Clues

EGDS was performed as part of the normal clinical practice and in accordance with international guidelines [3]. The examination was conducted at the referring Digestive Endoscopy Service of our hospital. The histological examination was performed by experienced CeD pathologists (more than 10 years of professional experience) according to internationally recognized criteria [3,16]. Histological findings were classified according to the classification proposed by Marsh and subsequently modified by Oberhuber [16]. The pathologist was unaware of the patient’s serological picture, as per our center’s clinical practice.

### 2.5. Statistical Analysis

Categorical variables were expressed as frequencies and percentages, and continuous variables were expressed as medians and interquartile ranges (IQRs). The difference between continuous variables was assessed with the Mann–Whitney test (2 groups) or Kruskal–Wallis test with Dunn’s post hoc correction (multiple groups). The frequency difference between the categorical variables was assessed with the chi-square test and Fisher’s exact test. Spearman’s rank correlation coefficient was used for linear regressions. A value of *p* < 0.05 was considered as statistically significant. The receiver operating characteristic (ROC) curve was analyzed and the area under the curve (AUC) was calculated to determine the accuracy of tTG levels at predicting Marsh 3 histological lesions (villous atrophy). ROC curve analysis was used to assess the accuracy of the binary test classification. Sensitivity, specificity, positive predictive value (PPV), and negative predictive value (NPV) were expressed as a percentage with a 95th percentile confidence interval. We divided the patients into three tertiles based on their total IgA levels at diagnosis, i.e., in patients with lower, intermediate, and higher total IgA values in the test population. We then assessed how the performance of the tTG-IgA/UNL ratio varied across different total IgA levels within these tertiles.

Statistical analysis was conducted with JASP software (Version 0.17.3) [Computer software], JASP Team (2023), Amsterdam, The Netherlands [17].

## 3. Results

### 3.1. Descriptive Statistics

A total of 165 patients satisfied the inclusion criteria. Patients’ demographic, clinical, and laboratory characteristics are summarized in Table 1.

### 3.2. Accuracy of Different tTG-IgA Cut-Offs in Predicting Villous Atrophy (Marsh 3)

Overall, Marsh 3 lesions were found in 142/165 patients (86.1%). The median tTG-IgA titer was 58.0 U/mL (24.0–128.0). The median tTG-IgA titer was 67.5 U/mL (30.25–128.0) in patients with evidence of intestinal villus atrophy (Marsh 3) and 24.0 U/mL (13.5–48.5) in patients without histological evidence of intestinal villi atrophy (Marsh 1 and 2), with a statistically significant difference (U 876.0, *p* < 0.001). A tTG-IgA value ≥ 10× UNL was found in 74/165 (44.8%) of patients. In patients with tTG-IgA ≥10 × UNL, intestinal villi atrophy (Marsh 3 histology) was found in 70/74 (94.6%) patients, while in patients with tTG-IgA <10× UNL, Marsh 3 histology was present in 72/91 (79.1%) patients, with a statistically significant difference (X = 8.146, *p* 0.004; Fisher’s exact test: Log Odds Ratio 1.522, *p* 0.006). Therefore, 4/74 (5.4%) of patients with tTG-IgA ≥ 10× UNL did not show the presence of intestinal villus atrophy (Marsh 3). Table 2 shows the accuracy of the tTG-IgA/UNL ratio with a threshold value of 10 x UNL in predicting the presence of Marsh 3 histological damage.

Analysis of the ROC curve (Figure 1) showed an AUC of 0.732 for tTG-IgA values in predicting villous atrophy with an optimal cut-off of tTG-IgA 6× UNL (42 U/mL) to maximize the diagnostic accuracy of the test. Table 2 shows the accuracy of the tTG-IgA/UNL ratio with a threshold value of 6 x UNL in predicting the presence of Marsh 3 histological damage.

In the study population, H. pylori-positive antral gastritis was incidentally found in 4/165 patients (2.4%) at EGDS. No neoplasia or other microscopic or macroscopic pathological findings were found incidentally. No patients required follow-up EGDS on suspicion of refractoriness.

### 3.3. Influence of the Levels of Total IgA on the Accuracy of Different tTG-IgA Cut Offs

Total IgA levels were known at the time of the first tTG-IgA determination in 130 out of 165 patients. As mentioned in the Methods, no total IgA values were considered when performed afterwards. Therefore, these analyses were performed on the 130 patients with available data.

The median total IgA value at diagnosis was 206.5 mg/dL IQR 156.25–263.25 mg/dL. Total IgA values were sex-related (F 192 mg/dL vs. M 235 mg/dL) (U 2257.5, *p* = 0.014), and age-related (Spearman’s rho 0.216, *p* 0.007). At a univariate linear regression, a linear correlation was found between tTG-IgA values and total IgA values (Spearman’s rho 0.165, *p* 0.030).

As indicated in the methods section, we divided the 130 patients into three tertiles based on the total IgA value at diagnosis: IgA < 174 mg/dL (43 patients), IgA ≥ 174 and <245 mg/dL (43 patients), and IgA ≥ 245 mg/dL (44 patients). The median tTG-IgA values at diagnosis in the three groups were, respectively, 56.0 mg/dL (IQR 22.5–128.0), 39.0 mg/dL (IQR 22.0–92.5), and 111.5 (IQR 45.5–128.0) The difference between median tTG IgA values at the diagnosis in the three groups was statistically significant H (2, n = 130) = 7.794, *p* = 0.020. Comparison of the three groups with Dunn’s post hoc correction showed a statistically significant difference between groups 2 and 3 and 1 and 3. We then tested how the performance of the tTG-IgA/UNL ratio could vary for different total IgA levels based on tertiles. Table 3 shows the change in the accuracy of the tTG-IgA/UNL ratio with a threshold value of 10× and 6× UNL in predicting the presence of Marsh 3 histological damage in patients with, respectively, total IgA < 174 mg/dL, ≥174 and <245 mg/dL, and ≥245 mg/dL.

Figure 2 shows the change in the binary classification of patients using a cut-off of tTG-IgA ≥ 10× and ≥6× UNL in the tree groups according to total IgA.

In the study population, 2 patients had IGAPD (IgA between 7 and 70 mg/dL) and 10 patients had increased total IgA (>400 mg/dL). The two patients with IGAPD had both Marsh 3 histology and tTG-IgA titer <10× UNL. In contrast, the 10 patients with IgA values >400 mg/dL had Marsh 3 histology in 9/10 and a tTG-IgA titer >10× UNL in 9/10. Of these patients, one patient had intestinal villi atrophy (Marsh 3) with a tTG-IgA titer <10× (false negative) and one patient had no intestinal villus atrophy with a tTG-IgA titer ≥10× (false positive).

### 3.4. Abnormal Levels of Total IgA

No correlation was found between the presence of normal, reduced, or increased IgA levels and age, sex, CeD-associated diseases, and symptoms. Total IgA > 400 mg/dL was also not more frequently associated with one or more autoimmune diseases. The only association found was between a IgA > 400 mg/dL and the symptom of diarrhea (X2 7.551, *p* = 0.006).

## 4. Discussion

In summary, using a tTG-IgA threshold of ≥10 times the UNL to predict intestinal villous atrophy (Marsh 3) showed sensitivity and specificity similar to previous studies in comparable populations (sensitivity: 50.0% vs. 54.0%, specificity: 77.8% vs. 90.0%) [6]. The threshold that best optimized diagnostic accuracy also matched earlier findings (>6 times UNL vs. 5.9 times UNL) [6]. Threshold values of tTG-IgA/UNL rates lower than 10× have also been proposed by other authors and depend on the population studied [5,18,19]. This could be explained by the lower tTG-IgA values found in adults compared to children [20]. Our data further confirmed that patients with higher levels of IgA antibodies tend to have higher total IgA values [13].

When diagnosing CeD using tTG-IgA tests, the choice of threshold values can significantly impact the results, especially depending on the total IgA levels of the patient. For patients with low or intermediate total IgA (below 245 mg/dL in our study), using a higher threshold (≥10× UNL) is very specific, but tends to miss some cases (lower sensitivity). However, if the threshold is lowered to ≥6× UNL, sensitivity increases while maintaining good specificity. In contrast, for patients with normal or high total IgA levels (above 245 mg/dL in our study), a higher threshold (≥10× UNL) proves to be quite effective, with high diagnostic accuracy. In this group, reducing the threshold does not provide additional benefits. In summary, adjusting the tTG-IgA threshold based on the patient’s total IgA levels can enhance diagnostic accuracy. For those with low or intermediate total IgA, a lower threshold improves detection rates, while for those with higher total IgA, a higher threshold remains effective and reliable. Future studies on the general population, and on a large sample, could identify a precise threshold value of total IgA that would make it more favorable to use the cutoff of tTG-IgA/UNL 10× instead of 6×.

Despite the small sample size, it appears challenging to apply a tTG-IgA/UNL threshold in patients with non-normal total IgA values. In patients with partial IgA deficiency (two patients), despite a tTG-IgA ratio <10× UNL (3.0 and 6.3), the histological lesion found was complete atrophy of the villi (Marsh 3c). In contrast, in patients with total IgA >400 mg/dL, the tTG-IgA cut-off ≥10× UNL was highly sensitive in predicting villous atrophy, but this may have been the group of patients with the highest false positive rate (specificity not calculable in the absence of true negative patients).

Patients with elevated total IgA (>400 mg/dL) did not exhibit a greater association with autoimmune diseases, as reported in pediatric patients [14]. However, they displayed a higher frequency of diarrhea, possibly indicative of increased intestinal inflammation.

Implementing a non-biopsy diagnosis of CeD through the tTG-IgA/UNL ratio did not lead to other occult gastrointestinal diseases being overlooked, excluding antral gastritis from H. pylori, which can be investigated using non-invasive tests (breath test or fecal antigen). No neoplastic pathology was detected.

Limitations of this study include its retrospective design, limited sample size, the lack of a control group, and a population with a high pre-test probability. The study specifically targeted patients who were already highly suspected of having CeD and had elevated tTG-IgA levels. This focus on a high-risk group was intended to directly evaluate the effectiveness of different threshold values in a context where the diagnosis was already considered likely, thus not requiring a control group for comparison. In this specific setting, the test had a high rate of false positives and a low rate of true negatives. Conversely, in a general population, using these thresholds of tTG-IgA/UNL could lead to higher accuracy, with higher specificity, in identifying those with the disease [7].

## 5. Conclusions

Despite the high specificity of a tTG-IgA threshold value of ≥10× UNL in predicting villous atrophy, the endorsement of a non-biopsy diagnosis of CeD in clinical practice is yet to be established by guidelines.

This study, to our knowledge, represents the first examination of the impact of total IgA values on the diagnostic accuracy of the tTG-IgA cut-offs in patients without selective IgA deficiency. While the general validity of the tTG-IgA ≥10× UNL cut-off is acknowledged, our study demonstrates how total IgA levels can influence optimal tTG-IgA cut-off values. While generalizing our conclusions to other populations might not be appropriate, the insights gained regarding the importance of total IgA values could help facilitate a better application of the non-biopsy diagnosis of CeD in the general adult population. This practical approach should be further addressed in a multicenter study to determine the optimal tTG-IgA/UNL threshold value based on total IgA titers, maximizing the diagnostic capacity of this test.

## Figures and Tables

**Figure 1 nutrients-16-03195-f001:**
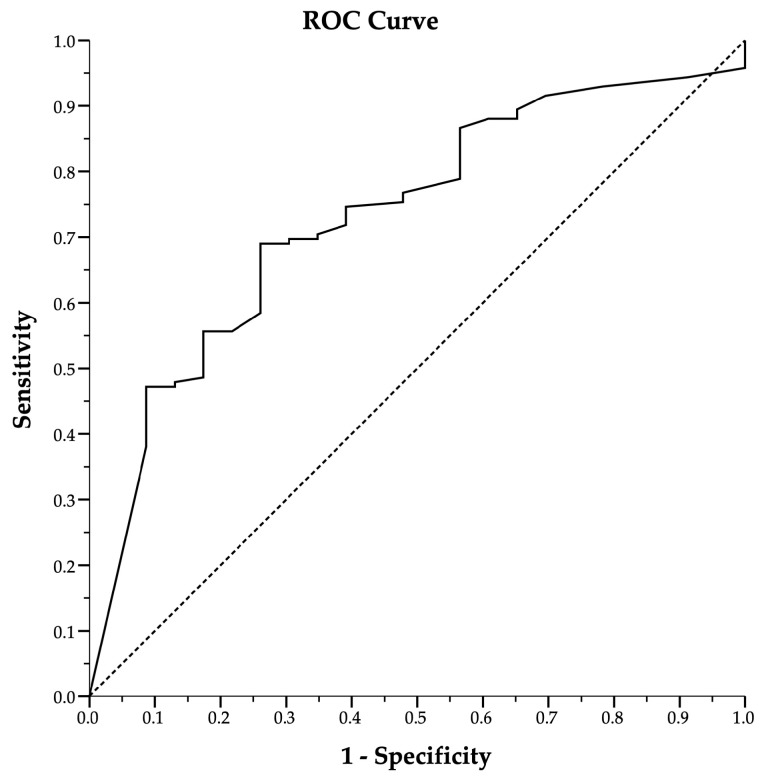
Receiver operating characteristic curve (ROC) analysis of tTG-IgA values against Marsh 3 histology. Diagonal dotted line represents the reference line (random classifier).

**Figure 2 nutrients-16-03195-f002:**
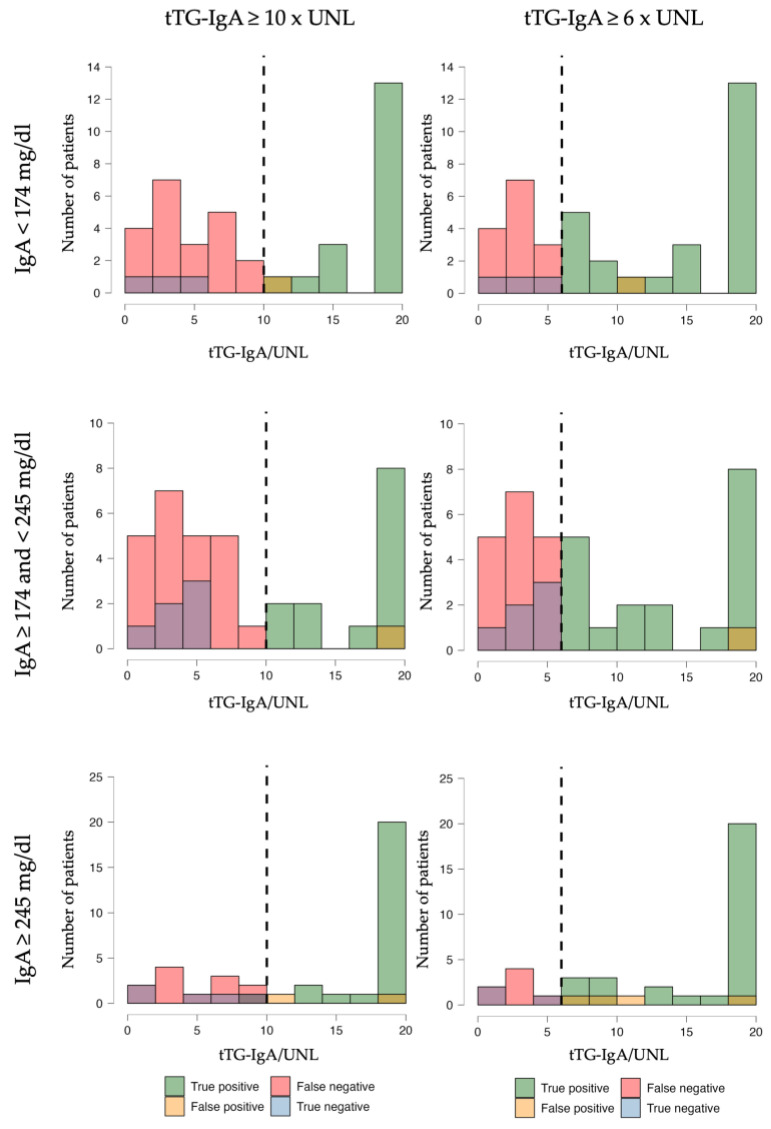
Signal detection plot of the binary classification of patients with and without Marsh 3 histological lesions according to an anti-trasglutaminase IgA antibodies (tTG-IgA) on upper normal level (UNL) rate >10× vs. >6× (dotted vertical lines) in patients with total IgA < 174 mg/dL (I tertile), total IgA ≥ 174 and <245 mg/dL (II tertile), and total IgA ≥ 245 mg/dL (III tertile). The colors shown in the legend may vary from the legend due to them overlapping within the graph.

**Table 1 nutrients-16-03195-t001:** Clinical and personal characteristics of the 165 patients included. tTG-IgA anti-transglutaminase IgA antibodies, IQR = interquartile range.

	Median	IQR	Min–Max
- Age (years)	42	27–54	18–80
- Total IgA (mg/dL)	206.5	156.25–263.25	50.0–1079.0
- tTG-IgA (U/mL)	58.0	24.0–128.0	7.0–128.0
	Number	%	
- Sex (F)	121/165	73%	
Histology			
- Marsh 1	14	8.5%	
- Marsh 2	9	5.5%	
- Marsh 3a	35	21.2%	
- Marsh 3b	62	37.6%	
- Marsh 3c	45	27.2%	
Clinical picture			
- Symptoms	124/165	75.2%	
- Classic symptoms	18/124	14.5%	
- Non-Classic symptoms	106/124	85.5%	
- Non-Classic Digestive	76/124	61.3%	
- Diarrhea	33/124	27%	
- Duhring dermatitis	5/165	3.0%	
- Screening	37/165	22.4%	
- Associated autoimmunity	48/165	29.1%	
- Family history	23/165	13.9%	

**Table 2 nutrients-16-03195-t002:** Diagnostic performance of a level of anti-transglutaminase IgA antibodies (tTG-IgA) ≥10 and ≥6 times the upper normal level (UNL) in predicting Marsh 3 histological lesions. 95% CI = 95% confidence interval.

tTG-IgA	≥10× UNL	95% CI	≥6× UNL	95% CI
Sensitivity	49.30%	40.81–57.81%	69.01%	60.72–76.50%
Specificity	82.61%	61.22–95.05%	73.91%	51.59–89.77%
PPV	94.59%	87.61–97.74%	94.23%	89.06–97.04%
NPV	20.88%	17.08–25.27%	27.87%	21.48–35.30%
Accuracy	53.94%	46.02–61.72%	69.70%	62.70–76.60%

**Table 3 nutrients-16-03195-t003:** Diagnostic performance of a level of anti-transglutaminase IgA antibodies (tTG-IgA) ≥10 vs. ≥6 times the upper normal level (UNL) in predicting Marsh 3 histological lesions in patients with total IgA < 174 mg/dL (I tertile), total IgA ≥ 174 and < 245 mg/dL (II tertile), and total IgA ≥ 245 mg/dL (III tertile).

	IgA < 174 mg/dL		IgA ≥ 174 and <245 mg/dL		IgA ≥ 245 mg/dL	
tTG-IgA	≥10× UNL	≥6× UNL	≥10× UNL	≥6× UNL	≥10× UNL	≥6× UNL
Sensitivity	46.15%	64.10%	36.11%	52.78%	67.57%	81.08%
Specificity	75.00%	75.00%	85.71%	85.71%	71.43%	42.86%
PPV	94.74%	96.15%	92.86%	95.00%	92.59%	88.24%
NPV	12.50%	17.65%	20.69%	26.09%	29.41%	30.00%
Accuracy	48.84%	65.12%	44.19%	58.14%	68.18%	75.00%

## Data Availability

The raw data supporting the conclusions of this article will be made available by the authors on request.

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
