# Peer review of "Exploring Total Immunoglobulin A’s Impact on Non-Biopsy Diagnosis of Celiac Disease: Implications for Diagnostic Accuracy"

_nutrients, 2024, doi:10.3390/nu16183195_

Round 1
Reviewer 1 Report
Comments and Suggestions for Authors
Line 41, tTG-IgA. Define the Abbreviation
Line 43, DQ-2 or DQ-8. Define the Abbreviation
Line 44, ESPGHAN. Define the Abbreviation
Lines 32-47. Merge in a single paragraph. Use simple sentences that readers can easily understand.
Lines 67-80. Merge in a single paragraph. Better to write the whole introduction in three comprehensive paragraphs with easy language.
Lines 89-96. Define demographics of the study population, gender, age, etc.
Lines 109-115. Merge in a single paragraph
Lines 173, 181. Move the sentences explaining methodology to the methods part.
In the results section, the titer plate readings result’ pictures are missing
Add more references to the discussion to strengthen the study results
Throughout the document, use simple sentences easy to understand. Also, throughout the document, merge all single sentences into proper paragraph structures.
The topic of the study is good, the research design can be further improved by adding study controls.
Without control the study results remain doubtful.
Comments on the Quality of English LanguageEnglish language of the article requires extensive editing in sentence and paragraph structure. The sentences used are complex and must be made simple and easy for readers.
Reviewer 2 Report
Comments and Suggestions for Authors
Dear Authors,
I read with great interest your amansucript about total IgA and coeliac disease.
You have many abbreviations in the text, please insert a list of abbreviations at the end of the manuscript.
In Table 1 is it Duhring dermatitis?
Line 152 - use capital letter for diagnostic.
Line 186 - use capital letter for diagnostic.
In Figure 1 insert specific values on all the colums.
Please format the references according to MDPI instructions, for example you have an article title written with capital letters.
I congratulate you on this extensive clinical study.
Reviewer 3 Report
Comments and Suggestions for Authors
Raiteri et al performed a retrospective single group study on untreated adults diagnosed with celiac disease with the gold standard method of duodenal endoscopic biopsies. The authors took advantage of the availability of baseline total IgA together with hTG IgA values in a good number of patients- the former routinely measured to verify that there is no IgA deficiency and, hence, the latter screening test is useful. In this study, the authors answered a different question: Do high total IgA values affect the performance of the hTG IgA test?
They showed that for patients with high total IgA, an increased cutoff value for hTG IgA would be optimal. Otherwise they verified poor sensitivity of the test, but high positive predictive value. The latter may set the grounds for no need for endoscopy and biopsy in the near future in hTG + individuals, but the study did not address this issue.
The manuscript is well-written and detailed data are provided. It would be of more clinical interest if the study was done on children, as this is the age group diagnosis can be done with serological markers alone- without endoscopy and biopsies. But, nevertheless, the submitted manuscript potentially merits publication, should all my major points below be addressed. I will be happy to review an updated version.
Major points
L168-72: Spearman's rhos of 0.216 and 0.165 denote very poor (if any) linear correlations. Please delete, rephrase and show why analysis was continued.
I feel that sensitivity and specificity analysis depicted with 6 bar graphs in figure 1 could be better shown with ROC curves, as usually done. I agree that the graphs provided give most of the primary data, but they are not illustrative enough.
Reviewer 4 Report
Comments and Suggestions for Authors
The authors assert: "The median tTG-IgA values at diagnosis in the 3 groups were respectively 56.0 mg/dl (IQR 22.5-128.0), 39.0 175 mg/dl (IQR 22.0-92.5) and 111.5 (IQR 45.5-128.0)" (row 174) - this suggests that the first two groups are not actually two distinct groups, at least from the tTG-IgA value point of view. Analyzing these two groups as separate entities is wrong from a biological point of view. Moreover, the authors should have understood that separating the whole group in three tertiles is artificial, as from a biological point of view there seem to exist only two categories of patients and the most adequate threshold between these two categories might not be 245 mg/dL. The authors should attempt to find this most adequate threshold: for which cut-off value for total IgA the difference in terms of tTG-IgA is the largest? Even better would be to find the cut-off value for total IgA for which changing the threshold from 10 x UNL to 6 x UNL does not result in a loss of specificity. If this value is not correctly identified, the practical value of the study is seriously undermined
Therefore, the conclusion of the study should be something like: "For patients with total IgA < [?] mg/dl, using a tTG-IgA cutoff of 6 x UNL instead of 10 x UNL maintained specificity ([?]%) with increased sensitivity (from [?]% to [?]%), while for patients with total IgA >= [?] mg/dl employing for tTG-IgA cutoff of 6 x UNL instead of 10 x UNL leads to an unacceptable loss in specificity."
[?] = to be determined
The authors assert "We then divided the 130 patients into 3 tertiles based on the total IgA value" - are we to understand that there were about 43-44 patients in each group?
Were the pathologists that examined the biopsies aware of the results of the blood tests (tTG-IgA, total IgA)?
What is the informative value of "At a univariate linear regression, a linear correlation was found between tTG-IgA 171 values and total IgA values (Spearman's rho 0.165, p 0.030)."? The correlation is actually surprisingly weak. tTG-IgA is a component of total IgA, therefore a positive correlation between the two is expected and consequently this result is trivial. Notwithstanding, the authors considered it sufficiently important to be mentioned in the Abstract.
There are unexplained abbreviations in the text, such as tTG-IgA.
Shonlein-Henoch -> Schönlein-Henoch (there are a "c" and an umlaut missing) (or, even better, Henoch-Schönlein)
În figure 1:
- What does "Marker" mean? - nothing is said about it in the legend. The authors should also clearly state what is the significance of the numbers on x axis, as well as of the dotted vertical line.
- The color used for True negative in the legend (light blue) is quite different from that used in the pictures (some kind of magenta).
Comments on the Quality of English LanguageThe formulation "Our data confirmed that in patients with higher levels of IgA antibodies, total IgA 216 values are higher" is a truism.
"resulted in 20 decreased specificity from 71.4% to 42.8% and increased sensitivity from 67.6% to 81.1%." should be reformulated as "resulted in a decrease in specificity from 71.4% to 42.8% and an increase in sensitivity from 67.6% to 81.1%."
The formulation "according to an anti-trasglutaminase IgA antibodies (tTG-IgA) on upper normal 194 level (UNL) rate 10 x vs. 6 x" is nonsensical in English.
Round 2
Reviewer 1 Report
Comments and Suggestions for Authors
The improved version of the manuscript is suitable now
Reviewer 3 Report
Comments and Suggestions for Authors
The authors adequately addressed my comments. So I recommend the manuscript for publication.